# Transfer Anomaly Detection by Inferring Latent Domain Representations

**Atsutoshi Kumagai**
NTT Software Innovation Center
NTT Secure Platform Laboratories
atsutoshi.kumagai.ht@hco.ntt.co.jp

**Tomoharu Iwata**
NTT Communication Science Laboratories
tomoharu.iwata.gy@hco.ntt.co.jp

**Yasuhiro Fujiwara**
NTT Communication Science Laboratories
yasuhiro.fujiwara.kh@hco.ntt.co.jp

## Abstract

We propose a method to improve the anomaly detection performance on target domains by transferring knowledge on related domains. Although anomaly labels are valuable to learn anomaly detectors, they are difficult to obtain due to their rarity. To alleviate this problem, existing methods use anomalous and normal instances in the related domains as well as target normal instances. These methods require training on each target domain. However, this requirement can be problematic in some situations due to the high computational cost of training. The proposed method can infer the anomaly detectors for target domains without re-training by introducing the concept of *latent domain vectors*, which are latent representations of the domains and are used for inferring the anomaly detectors. The latent domain vector for each domain is inferred from the set of normal instances in the domain. The anomaly score function for each domain is modeled on the basis of autoencoders, and its domain-specific property is controlled by the latent domain vector. The anomaly score function for each domain is trained so that the scores of normal instances become low and the scores of anomalies become higher than those of the normal instances, while considering the uncertainty of the latent domain vectors. When target normal instances can be used during training, the proposed method can also use them for training in a unified framework. The effectiveness of the proposed method is demonstrated through experiments using one synthetic and four real-world datasets. Especially, the proposed method without re-training outperforms existing methods with target specific training.

## 1 Introduction

Anomaly detection is an important task in artificial intelligence [8, 6]. The goal of anomaly detection is to detect anomalous instances, called anomalies or outliers, that do not conform to the expected normal pattern. Anomaly detection methods have been used in a wide variety of applications such as intrusion detection [16], fraud detection [33], medical care [30], and industrial asset monitoring [28].

Many semi-supervised anomaly detection methods have been proposed such as autoencoders (AEs) [43], one-class support vector machines (OSVMs) [44], and isolation forests [35]. Since they require only normal instances, which are relatively easy to prepare, to obtain anomaly detectors, they are particularly used in practice. In some situations, anomaly labels that indicate anomalous instances can be used. By using anomaly labels, supervised anomaly detection methods can detect anomalies

much better than semi-supervised ones [29, 50, 23, 10]. Although anomaly labels are valuable, they are typically difficult to obtain since anomalies rarely occur.

Even if anomaly labels are difficult to obtain in a domain of interest, called a target domain, they may be obtainable in related domains, called source domains. For example, in cyber-security, security companies monitor customers' networks to prevent cyber-attacks [14]. Although anomalies are difficult to obtain from a new customer's network (target domain), they may be obtained from existing customers' networks (source domains) that have long been monitored. Similarly, in condition-based monitoring of industrial assets such as coal mining drilling machines using sensor data [27], although anomalies are difficult to obtain from a new asset, anomalies may be obtained from related assets that have long been working.

Several transfer anomaly detection methods have been proposed to learn the anomaly detector by using both anomalous and normal instances in the source domains [3, 7, 28, 17, 48]. These methods also use target normal instances for training. However, training after obtaining target instances can be problematic in some applications. For example, consider anomaly detection on Internet-of-Things (IoT) devices such as sensors, cameras, and cars. Since each device does not have sufficient computational resources, training is difficult to perform on these devices even if target domains appear that contain normal instances. As another example, in cyber-security, widely various IoT devices themselves need to be protected from cyber attacks [4]. However, it is difficult to protect all devices quickly with time-consuming training since many new devices (target domains) appear one after another.

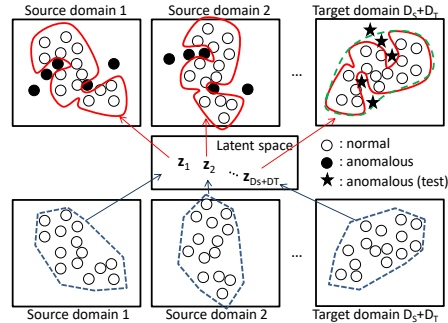

In this paper, we propose a novel method to improve the anomaly detection performance on target domains by using anomalous and normal instances, or only normal instances in source domains. The proposed method can infer the anomaly detectors for any domains given the sets of normal instances in the domains without re-training. In addition, the proposed method can use target normal instances for training in a unified framework when they can be used during training. With the proposed method, an anomaly score function, which outputs an anomaly score given an instance, is defined on the basis of the AE, which is widely used in recent anomaly detection methods [43, 56, 12, 1, 50]. Note that the proposed method can use other semi-supervised anomaly detection methods with learnable parameters instead of AEs such as variational AEs [32] and energy-based models [55]. The parameters of the AE are shared across all domains to learn the common property of domains. However, each domain has a specific property that cannot be explained only by the common property. To reflect these domain-specific properties to the anomaly score function efficiently, we intro-

Figure 1: The latent domain vector $\mathbf{z}_d$ is inferred from the set of normal instances in the domain (blue dotted line) so as to detect anomalies. The decision boundary for each domain (red line) is induced by the domain-specific anomaly score function, which is determined by $\mathbf{z}_d$. The green dotted line denotes the decision boundary of AE, which uses only target normal instances. The proposed method can detect test anomalies that AE cannot via data on the related domains.

duce *latent domain vectors*, which are latent representations of domains. The latent domain vectors are used to condition the anomaly score function and control the property of the anomaly score function for each domain.

The latent domain vector of each domain is estimated from the set of normal instances in the domain by a neural network. This model enables us to infer the anomaly detectors for any domains given the sets of normal instances in the domains without re-training or anomalies. To infer the latent domain vectors of different domains, the neural networks need to take sets with different sizes of normal instances as inputs. We realize this by using the deep sets [54], which are permutation invariant neural networks to the order of instances in the sets. Specifically, we model the parameters for the posterior of the latent domain vector with the deep sets.

The anomaly score function for each domain is trained so that the anomaly scores of normal instances become low, which is achieved by minimizing the reconstruction error of the normal instances. Anomaly labels are used so that the anomaly scores of anomalous instances become higher than those

of normal instances, which is realized by adding a differentiable area under the curve (AUC) loss as the regularizer. This regularizer enables us to improve the performance even if the number of anomalies is small [29]. Since the latent domain vectors are estimated from data, the estimation is often uncertain. To handle this uncertainty appropriately, we take the expectation of the loss function w.r.t. the latent domain vector and regularize the posterior of the latent domain vector by a prior using Kullback Leibler (KL) divergence, which is used as our objective function for the domain. The parameters for the anomaly score function and the posteriors of the latent domain vectors are estimated simultaneously by minimizing the sum of the objective function for each domain. Figure 1 illustrates the proposed method.

## 2 Related Work

Anomaly detection, which is also called outlier detection or novelty detection, has been widely studied [8, 6]. Although many unsupervised or semi-supervised anomaly detection methods have been proposed such as AE based methods [43, 56, 12, 1], OSVM based methods [44, 7, 40], and density based methods [57, 51, 55], they cannot use anomaly labels. Supervised anomaly detection uses anomaly labels to improve the performance, which is also referred to as imbalanced classification [9, 21]. They assume that all instances are obtained from one domain and thus cannot perform well when there is a domain difference which is focused on in this paper.

Transfer learning or domain adaptation aims to solve a problem in a target domain using data in source domains [39]. Unsupervised approaches aim to adapt to the target domain by using labeled source data and target unlabeled data [26, 37, 5, 42, 36, 34]. Semi-supervised approaches use labeled target data as well as these data [38, 13, 41]. Although these methods performed impressively, they usually do not assume the class-imbalance and thus are not appropriate for anomaly detection [47]. Although several transfer learning methods are designed to deal with the class-imbalance [2, 53, 18], they assume anomalous and normal instances in the target domains for training. The proposed method does not assume target anomalous instances, which is more practical since anomalies rarely occur.

Some transfer anomaly detection methods do not use target anomalies to obtain anomaly detectors as with the proposed method. The two-step approach is widely used in transfer anomaly detection [3, 7]. This approach first extracts discriminative features from data in the source domain with neural networks. After feature extraction, any semi-supervised algorithms like OSVM are applied to the target normal instances. Although this approach is effective to some extent, dividing anomaly detector and feature (transfer) learning risks losing information to be transferred [7, 40]. In contrast, the proposed method learns them simultaneously in an end-to-end learning manner. A few methods use only normal instances in both the source and target domains [28, 17, 48]. One method uses an auxiliary dataset of outliers [22]. Another assumes target unlabeled data for training [47]. All these methods require training on each target domain, which can be problematic in some situations as described in Section 1. The proposed method can instantly infer the anomaly detector for each domain given the set of normal instances in the domain without re-training.

One-class data transfer learning (OTL) [11] can predict classifiers for new domains without re-training although this ability is not mentioned by Chen and Liu [11]. OTL trains a regression model that predicts parameters of anomaly density from those of a normal density using labeled source data. OTL predicts a target classifier by predicting densities given target normal instances. Although OTL requires the anomaly density to be estimated for each source domain, this is quite difficult since the number of anomalies is small in anomaly detection tasks. In addition, OTL cannot use domains that contain only normal instances for training although the proposed method can.

## 3 Preliminary

AEs are neural networks originally proposed for non-linear dimensionality reduction [25]. Due to their simplicity and effectiveness, AEs have become a fundamental component of recent semi-supervised anomaly detection [6, 43, 56, 12, 1]. Thus, we use AEs as a building block of the proposed method. Given instances $\mathbf{X} := \{\mathbf{x}_1, \ldots, \mathbf{x}_N\}$, the AE is trained by minimizing the following loss function, $L(\theta_F, \theta_G) := \frac{1}{N} \sum_{n=1}^{N} \|\mathbf{x}_n - G_{\theta_G}(F_{\theta_F}(\mathbf{x}_n))\|^2$, where $F_{\theta_F}$ is a neural network with the parameter $\theta_F$, called the encoder; $G_{\theta_G}$ is a neural network with the parameter $\theta_G$, called the decoder; $\|\cdot\|$ is a Euclidean norm; and $\|\mathbf{x} - G_{\theta_G}(F_{\theta_F}(\mathbf{x}))\|^2$ is a reconstruction error of $\mathbf{x}$. When the AE is

trained with normal instances, the reconstruction errors of normal instances become low. In contrast, the reconstruction errors of instances dissimilar to normal instances, i.e., anomalies, can be expected to become high since they are not learned. Thus, AEs can be used for anomaly detection with the reconstruction error as the anomaly score.

# 4 Proposed Method

## 4.1 Task

Let $\mathbf{X}_d^+ := \{\mathbf{x}_{dn}^+\}_{n=1}^{N_d^+}$ be a set of anomalous instances in the $d$-th domain, where $\mathbf{x}_{dn}^+ \in \mathbb{R}^M$ is the $M$-dimensional feature vector of the $n$-th anomalous instance of the $d$-th domain and $N_d^+$ is the number of the anomalous instances in the $d$-th domain. Similarly, let $\mathbf{X}_d^- := \{\mathbf{x}_{dn}^-\}_{n=1}^{N_d^-}$ be a set of normal instances in the $d$-th domain. We assume that $N_d^+ \ll N_d^-$ in each domain since anomalies rarely occur, and the feature vector size $M$ is the same in all domains as in many existing studies [38, 20, 28, 42]. Suppose that we have both anomalous and normal instances in $D_\mathrm{S}$ source domains, $\{\mathbf{X}_d^+ \cup \mathbf{X}_d^-\}_{d=1}^{D_\mathrm{S}}$, and normal instances in $D_\mathrm{T}$ target domains, $\{\mathbf{X}_d^-\}_{d=D_\mathrm{S}+1}^{D_\mathrm{S}+D_\mathrm{T}}$. Note that the proposed method can also treat source domains that have only normal instances although we assumed that all source domains have anomalies for simplicity. Our goal is to obtain an appropriate domain-specific anomaly score function, which outputs its anomaly score given an instance, for each target domain.

## 4.2 Domain-specific Anomaly Score Function

We define the domain-specific anomaly score function based on the reconstruction error of the AE. To represent the property of each domain efficiently, we assume that each domain has a $K$-dimensional latent continuous variable $\mathbf{z}_d \in \mathbb{R}^K$, which is called a latent domain vector in this paper. For the $d$-th domain, we define the anomaly score function conditioned on the latent domain vector $\mathbf{z}_d$ as follows,

$$s_\theta(\mathbf{x}_{dn}|\mathbf{z}_d) := \|\mathbf{x}_{dn} - G_{\theta_G}(F_{\theta_F}(\mathbf{x}_{dn}, \mathbf{z}_d))\|^2, \tag{1}$$

where the parameters $\theta := (\theta_F, \theta_G)$ are shared among all domains. Unlike the original reconstruction error, this score function depends on the latent domain vector $\mathbf{z}_d$. By changing the value of $\mathbf{z}_d$, the proposed method can flexibly control the property of the anomaly score function. Although we use the reconstruction error for simplicity, we can use other anomaly score functions with learnable parameters such as autoregressive density models [19, 46] and flow-based density models [15, 31].

## 4.3 Models for Latent Domain Vectors

Since the latent domain vectors are unobserved, we have to estimate them from data. For a first step, we model the conditional probability of the latent domain vector given the set of normal instances as a multivariate Gaussian distribution with a diagonal covariance matrix:

$$q_\phi(\mathbf{z}_d|\mathbf{X}_d^-) := \mathcal{N}(\mathbf{z}_d|\mu_\phi(\mathbf{X}_d^-), \mathrm{diag}(\sigma_\phi^2(\mathbf{X}_d^-))), \tag{2}$$

where mean $\mu_\phi(\mathbf{X}_d^-) \in \mathbb{R}^K$ and variance $\sigma_\phi^2(\mathbf{X}_d^-) \in \mathbb{R}_+^K$ are modeled by neural networks with parameters $\phi$ that are shared among all domains, and $\mathrm{diag}(\mathbf{x})$ returns a diagonal matrix whose diagonal elements are $\mathbf{x}$. In this model, the latent domain vector $\mathbf{z}_d$ depends on the set of normal instances $\mathbf{X}_d^-$. By this modeling, we can infer the latent domain vectors of any domains when the sets of normal instances in the domains are given. Accordingly, we can obtain domain-specific anomaly score functions for the domains without re-training or anomalies.

Since the $q_\phi$ deals with the set of normal instances $\mathbf{X}_d^-$ as an input, the neural networks for the parameters $\mu_\phi(\mathbf{X}_d^-)$ and $\ln \sigma_\phi^2(\mathbf{X}_d^-) \in \mathbb{R}^K$ must be permutation invariant to the order of instances in the set. To achieve this, we use the recently proposed deep sets architecture [54], $\tau(\mathbf{X}_d^-) = \rho\left(\sum_{n=1}^{N_d^-} \eta(\mathbf{x}_{dn}^-)\right)$, where $\tau(\mathbf{X}_d^-)$ represents one of the $\mu_\phi(\mathbf{X}_d^-)$ and $\ln \sigma_\phi^2(\mathbf{X}_d^-)$, $\rho$ and $\eta$ are any neural networks, respectively. This architecture is permutation invariant due to the summation. Although this architecture is quite simple, it can express any permutation invariant function and preserve all the properties of the set with suitable $\rho$ and $\eta$ [54]. Thus, we can capture the property of each domain well with this architecture.

## 4.4 Objective Function

We define the objective function of the proposed method using the domain-specific anomaly score functions and latent domain vectors. First, the objective function for the $d$-th source domain conditioned on the latent domain vector $\mathbf{z}_d$ to be minimized is defined by

$$L_d(\theta|\mathbf{z}_d) := \frac{1}{N_d^-} \sum_{n=1}^{N_d^-} s_\theta(\mathbf{x}_{dn}^-|\mathbf{z}_d) - \frac{\lambda}{N_d^- N_d^+} \sum_{n,m=1}^{N_d^-,N_d^+} f(s_\theta(\mathbf{x}_{dm}^+|\mathbf{z}_d) - s_\theta(\mathbf{x}_{dn}^-|\mathbf{z}_d)), \qquad (3)$$

where $\lambda \geq 0$ is the hyperparameter and $f$ is the sigmoid function, $f(x) = \frac{1}{1+\exp(-x)}$. This form of the objective function has recently been proposed by Iwata and Yamanaka [29] and showed better performance than existing methods although they do not consider domain differences. The first term of Eq. (3) represents the anomaly scores of normal instances in the $d$-th domain. Since the anomaly scores of normal instances should be low, we minimize this term. The second term of Eq. (3) is a differentiable approximation of the AUC [52], which is effective for class-imbalanced data [23]. The anomaly scores of anomalous instances should be higher than those of normal instances, $s_\theta(\mathbf{x}_{dm}^+|\mathbf{z}_d) > s_\theta(\mathbf{x}_{dn}^-|\mathbf{z}_d)$ for any $\mathbf{x}_{dm}^+ \in \mathbf{X}_d^+, \mathbf{x}_{dn}^- \in \mathbf{X}_d^-$. The AUC term encourages this since the $f(\cdot)$ takes the maximal value one when $s_\theta(\mathbf{x}_{dm}^+|\mathbf{z}_d) \gg s_\theta(\mathbf{x}_{dn}^-|\mathbf{z}_d)$ and the minimal value zero when $s_\theta(\mathbf{x}_{dm}^+|\mathbf{z}_d) \ll s_\theta(\mathbf{x}_{dn}^-|\mathbf{z}_d)$. When there are no anomalies or $\lambda = 0$, the second term of Eq. (3) becomes zero and the first term of Eq. (3) remains. Thus, it is a supervised extension of the AE described in Section 3.

Since the latent domain vector $\mathbf{z}_d$ has uncertainty with the variance $\sigma_\phi^2$, we want to appropriately take this into account for the objective function. To achieve this, we define the objective function for the $d$-th source domain to be minimized as follows

$$\mathcal{L}_d(\theta, \phi) := \mathbb{E}_{q_\phi(\mathbf{z}_d|\mathbf{X}_d^-)}[L_d(\theta|\mathbf{z}_d)] + \beta D_{\mathrm{KL}}(q_\phi(\mathbf{z}_d|\mathbf{X}_d^-)||p(\mathbf{z}_d)), \qquad (4)$$

where $D_{\mathrm{KL}}(q_\phi(\mathbf{z}_d|\mathbf{X}_d^-)||p(\mathbf{z}_d))$ is the KL divergence between $q_\phi(\mathbf{z}_d|\mathbf{X}_d^-)$ and a standard Gaussian distribution $p(\mathbf{z}_d) := \mathcal{N}(\mathbf{0}, \mathbf{I})$, and $\beta > 0$ is a hyperparameter. The first term of Eq. (4) is the expectation of the objective function (3) w.r.t. the $q_\phi(\mathbf{z}_d|\mathbf{X}_d^-)$. Since the expectation considers all the probabilities of the $\mathbf{z}_d$, it can lead to robust training. This expectation term can be effectively approximated by the reparametrization trick [32]. That is, $\mathbb{E}_{q_\phi(\mathbf{z}_d|\mathbf{X}_d^-)}[L_d(\theta|\mathbf{z}_d)] \approx \frac{1}{L} \sum_{\ell=1}^{L} L_d(\theta|\mathbf{z}_d^{(\ell)})$, where $\mathbf{z}_d^{(\ell)} = \mu_\phi(\mathbf{X}_d^-) + \epsilon_d^{(\ell)} \odot \sigma_\phi(\mathbf{X}_d^-)$, $\epsilon_d^{(\ell)} \sim \mathcal{N}(\mathbf{0}, \mathbf{I})$, and $\odot$ is an element-wise product. The second term of Eq. (4) is a regularization term that prevents over-fitting of the latent domain vectors, where its strength is controlled by $\beta$. This type of regularization is common in variational AEs [32, 24] and can be analytically calculated [32]. The $q_\phi(\mathbf{z}_d|\mathbf{X}_d^-)$ is trained so as to minimize the loss term (the first term of Eq. (4)) while being constrained to the prior $p(\mathbf{z}_d)$. The effectiveness of considering the uncertainty will be demonstrated in our experiments.

The objective function for the $d$-th target domain to be minimized is obtained by omitting the AUC loss term from Eq. (4) since the target domain does not have anomalous instances. That is,

$$\mathcal{L}_d(\theta, \phi) := \mathbb{E}_{q_\phi(\mathbf{z}_d|\mathbf{X}_d^-)} \left[ \frac{1}{N_d^-} \sum_{n=1}^{N_d^-} s_\theta(\mathbf{x}_{dn}^-|\mathbf{z}_d) \right] + \beta D_{\mathrm{KL}}(q_\phi(\mathbf{z}_d|\mathbf{X}_d^-)||p(\mathbf{z}_d)), \qquad (5)$$

where the first term can also be approximated using the reparametrization trick. As a result, the objective function for the proposed method is the following weighted sum of the objective function for each domain, $\mathcal{L}(\theta, \phi) := \sum_{d=1}^{D_\mathrm{S}+D_\mathrm{T}} \alpha_d \mathcal{L}_d(\theta, \phi)$, where $\alpha_d \geq 0$ is the hyperparameter. This objective function can be minimized w.r.t. $\theta$ and $\phi$ by gradient-based optimization methods. This formulation includes various settings. For example, when no target instances can be used in the training phase, we set $\alpha_d = 0$ for $d = D_\mathrm{S} + 1, \ldots, D_\mathrm{S} + D_\mathrm{T}$ and $\alpha_d = 1$ for $d = 1, \ldots, D_\mathrm{S}$. The proposed method can infer the anomaly score functions for the domains that are not used for training when the sets of normal instances in the domains are given without re-training as described below.

## 4.5 Inference

By using the learned parameters $(\theta_*, \phi_*)$ and the normal instances $\mathbf{X}_{d'}^-$, the proposed method infers the domain-specific anomaly score function for the $d'$-th domain as follows:

$$s(\mathbf{x}_{d'}) := \int s_{\theta_*}(\mathbf{x}_{d'}|\mathbf{z}_{d'})q_{\phi_*}(\mathbf{z}_{d'}|\mathbf{X}_{d'}^-)d\mathbf{z}_{d'} \approx \frac{1}{L}\sum_{\ell=1}^{L} s_{\theta_*}(\mathbf{x}_{d'}|\mathbf{z}_{d'}^{(\ell)}) \tag{6}$$

where $\mathbf{z}_{d'}^{(\ell)} = \mu_{\phi_*}(\mathbf{X}_{d'}^-) + \epsilon^{(\ell)} \odot \sigma_{\phi_*}(\mathbf{X}_{d'}^-)$, $\epsilon^{(\ell)} \sim \mathcal{N}(\mathbf{0}, \mathbf{I})$, and $\mathbf{x}_{d'}$ is any instance in the $d'$-th domain. The proposed method can infer the $s(\cdot)$ considering the uncertainty of the latent domain vectors by sampling $\mathbf{z}_{d'}$ from the $q_{\phi_*}(\mathbf{z}_{d'}|\mathbf{X}_{d'}^-)$, which enables robust anomaly detection. This can be inferred even if $\mathbf{X}_{d'}^-$ are not used for training. The computational complexity for the inference is $\mathcal{O}(N_{d'}^- + L)$.

## 5 Experiments

We demonstrate the effectiveness of the proposed method using one synthetic and four real-world class-imbalanced datasets. To measure anomaly detection ability on target domains, we evaluated the AUC, which is a well used measure for anomaly detection tasks, on one domain while training the rest. We used the following setup: CPU was Intel Xeon E5-2660v3 2.6 GHz, the memory size was 128 GB, and GPU was NVIDIA Tesla k80.

### 5.1 Data

We created the simple two-dimensional dataset shown in Figure 2a. This dataset consists of eight double circles (domains) around the $(0,0)$. Each double circle has outer and inner circles that consist of normal and anomalous instances, respectively. We used the '7'-th domain as the target domain and the rest as the source domains. We used four real-world public datasets: MNIST-r, Anuran Calls, Landmine, and IoT. The MNIST-r is derived from the MNIST by rotating images, which was introduced by Ghifary et al. [20]. The MNIST-r has six domains. We selected the '4' digit as the anomalous class and the rest as the normal class since it was the most difficult setting in our preliminarily experiment. The Anuran Calls is a real-world dataset collected from frog croaking sounds, which is used in the multi-task anomaly detection study [28]. We regarded each specie as a domain referring to the study [28], and thus, the Anuran Calls has five domains. The Landmine is a real-world dataset that is well-used in multi-task learning [49]. We used ten domains that consist of the first five (1-5) and last five (25-29) domains. The IoT contains real network traffic data, which are gathered from nine IoT devices infected by BASHLITE malware. We did not use the device that had no normal data, and thus, the IoT has eight domains. The average anomaly rate $N_d^+/(N_d^+ + N_d^-)$ of Synthetic, MNIST-r, Anuran Calls, Landmine, and IoT are 0.048, 0.1, 0.024, 0.062, and 0.05, respectively. Due to the length limit of the paper, the details of the datasets including download links are provided in the supplemental material.

### 5.2 Comparison Methods

We evaluated two variants of the proposed method: ProT and ProS. ProT uses normal instances in the target domain as well as anomalous and normal source instances for training. ProS does not use target normal instances for training. After training with source domains, ProS infers the anomaly score function for the target domain using the set of target normal instances without re-training. The proposed method was implemented by Chainer [45].

We compared the proposed method with eight methods: the feed-forward neural network classifier (NN), the NN for class-imbalanced data (NNAUC), the autoencoder based classifier for class-imbalanced data (AEAUC) [29], the autoencoder (AE) [43], the one-class support vector machine (OSVM) [44], the contrastive semantic alignment (CCSA) [38], the transfer one-class support vector machine (TOSVM) [3], and the one-class data transfer learning (OTL) [11]. AE and OSVM are semi-supervised anomaly detection methods, which use only normal instances in the target domain for training. NN, NNAUC, and AEAUC are supervised anomaly detection methods, which use both anomalous and normal instances in the source domains for training. AEAUC is obtained from ProS by omitting the latent domain vectors. CCSA, TOSVM, and OTL are transfer learning or transfer anomaly detection methods, which use both anomalous and normal instances in the source domains

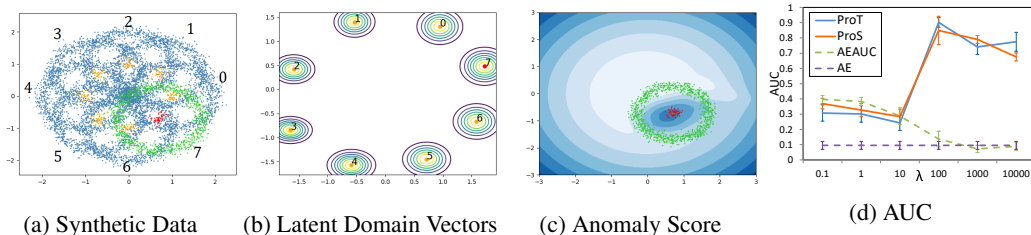

| (a) Synthetic Data | (b) Latent Domain Vectors | (c) Anomaly Score | (d) AUC |

Figure 2: (a) Synthetic dataset consists of eight double circles (domains). Anomalous and normal source instances are represented by orange and blue points, respectively. Anomalous and normal target instances are represented by red and green points, respectively. (b) Posteriors of the latent domain vectors estimated by ProS. Orange points denote the mean of each posterior. (c) Heatmap of anomaly scores on the target domain obtained by ProS. Darker color indicates higher anomaly score. (d) Average and standard error of AUCs when $\lambda$ was changed.

Table 1: Average and standard deviation of AUCs [%] on each target domain on MNIST-r.

| Target | ProT | ProS | NN | NNAUC | AEAUC | AE | OSVM | CCSA | TOSVM | OTL |
|---|---|---|---|---|---|---|---|---|---|---|
| 0 | **93.9(0.7)** | 92.5(1.2) | 88.7(1.2) | 87.2(1.3) | 88.5(2.8) | 72.6(2.5) | 56.7(1.2) | 87.3(1.3) | 89.4(4.8) | 86.5(1.3) |
| 15 | **99.2(0.3)** | **99.2(0.3)** | 98.2(0.4) | 97.7(0.3) | 98.7(0.5) | 72.6(2.5) | 63.9(1.2) | 97.7(0.4) | 95.5(1.8) | 95.2(0.8) |
| 30 | **98.8(0.4)** | **98.8(0.4)** | 98.2(0.3) | 97.6(0.4) | 98.1(0.6) | 71.8(3.9) | 63.5(1.3) | 97.5(0.4) | 94.6(2.4) | 95.3(1.0) |
| 45 | **97.1(0.8)** | 95.3(0.8) | **96.4(0.8)** | 94.7(0.9) | 94.6(1.6) | 73.6(3.6) | 63.9(0.9) | **96.5(1.0)** | 90.6(4.8) | 91.9(1.4) |
| 60 | **99.2(0.4)** | 99.2(0.2) | 98.9(0.5) | 98.7(0.3) | 98.2(0.9) | 72.8(2.3) | 63.6(1.0) | 98.2(0.5) | 96.2(1.8) | 94.8(1.4) |
| 75 | **91.2(2.0)** | 91.1(2.9) | 88.7(2.2) | 88.5(1.4) | 87.7(2.3) | 72.9(2.9) | 71.4(2.2) | **92.3(0.8)** | 93.5(2.8) | 78.4(3.0) |
| Avg | **96.6(3.2)** | 96.0(3.6) | 94.8(4.6) | 94.1(4.7) | 94.3(4.9) | 72.7(3.0) | 63.8(4.5) | 94.9(4.1) | 93.3(4.1) | 90.3(6.4) |

and normal instances in the target domain to obtain the anomaly detector. Although CCSA and TOSVM use target normal instances for training like ProT, OTL does not like ProS.

We selected hyperparameters using average validation AUC on the source domains for all methods except for AE. AE used the validation reconstruction error on the target domains. We evaluated the test AUC when the method obtained the best validation AUC after 15 epochs to avoid over-fitting. We conducted experiments on ten different datasets for each target domain and reported the mean test AUC. The details of the experimental setup such as network architectures for the proposed method and comparison methods and hyperparameter candidates are described in the supplemental material.

## 5.3 Results

First, we show how the proposed method works by using the synthetic dataset. Each domain of the synthetic dataset is located on a circle as shown Figure 2a. Figure 2b represents an example of the posterior of the latent domain vectors estimated by ProS with $K = 2$. We found that the estimated latent domain vectors were located on a circle to preserve the relationship between the domains. Especially, the posterior of the target domain ('7'-th domain) was predicted between the posteriors of the '0'-th and '6'-th domains even if the target domain was not used for training (i.e., the objective function). The parameters of $q_\phi(\mathbf{z}_d|\mathbf{X}_d^-)$ are inferred from an input $\mathbf{X}_d^-$ so as to find anomalies. Since the parameters of $q_\phi(\mathbf{z}_d|\mathbf{X}_d^-)$ are continuous w.r.t. the input $\mathbf{X}_d^-$, the posteriors of $\mathbf{z}_d$'s become similar if the $\mathbf{X}_d^-$'s are similar. Thus, similar domains (e.g., the '3'-th and '4'-th domains) were located near each other in the latent space. Figure 2c represents an example of the heatmap of anomaly scores on the target domain obtained by ProS. We found that ProS was able to give high (low) anomaly scores to the anomalous (normal) region on the target domain, i.e., AUC is one even if the target domain was not used for training. Figure 2d shows average and standard error of AUCs with different $\lambda$ on the target domain. We used AEAUC as a baseline since it is obtained from ProS by omitting the latent domain vectors. ProT and ProS outperformed the others by large margins when the values of $\lambda$ were relatively large. This result indicates the importance of both anomalies and modeling domain differences. Overall, these results demonstrate that the proposed method can capture the property of domains as the latent domain vectors and perform well.

Second, we evaluated anomaly detection performance on the target domain using the real-world datasets. Tables 1 – 4 show the average and standard deviation of AUCs with different target domains

Table 2: Average and standard deviation of AUCs [%] on each target domain on Anuran Calls.

| Target | ProT | ProS | NN | NNAUC | AEAUC | AE | OSVM | CCSA | TOSVM | OTL |
|---|---|---|---|---|---|---|---|---|---|---|
| Ade | **99.9(0.1)** | 95.5(4.8) | 84.8(19.) | 64.0(17.) | 95.6(4.0) | 84.5(12.) | 96.6(0.9) | 95.2(4.4) | 77.9(20.) | 90.9(4.7) |
| Ame | **99.4(1.3)** | 93.5(3.6) | 87.1(4.5) | 85.7(5.2) | 94.1(5.5) | 81.6(8.1) | 89.1(3.0) | 96.2(2.6) | 93.6(5.7) | 93.8(2.9) |
| Hyl | **99.7(0.3)** | 98.6(2.0) | **99.6(0.9)** | 99.4(1.0) | **98.9(2.0)** | 93.8(3.7) | 89.9(1.7) | **99.6(0.4)** | **95.4(6.3)** |
| HCi | **99.9(0.2)** | 98.9(1.6) | 98.7(2.1) | 98.2(3.1) | **99.4(0.8)** | 89.5(1.2) | 93.6(0.6) | 96.7(2.8) | **99.3(1.3)** | 90.4(4.5) |
| HCo | **99.9(0.1)** | 97.6(3.5) | 97.1(4.7) | 96.2(3.6) | 96.5(3.4) | 94.2(6.3) | 92.7(1.7) | **99.5(1.2)** | 97.4(1.7) | 97.2(2.4) |
| Avg | **99.8(0.6)** | 96.8(3.8) | 93.4(11.) | 88.7(16.) | 96.9(4.0) | 88.7(8.5) | 92.4(3.2) | 96.7(4.1) | 93.6(12.) | 93.5(4.9) |

Table 3: Average and standard deviation of AUCs [%] on each target domain on Landmine.

| Target | ProT | ProS | NN | NNAUC | AEAUC | AE | OSVM | CCSA | TOSVM | OTL |
|---|---|---|---|---|---|---|---|---|---|---|
| 1 | **83.9(2.5)** | **83.8(2.0)** | 77.8(2.0) | 80.9(2.8) | 78.8(3.3) | 50.0(5.7) | 38.0(7.2) | 78.2(3.4) | 55.9(6.2) | 61.0(6.2) |
| 2 | **78.4(2.2)** | **78.5(2.3)** | 73.8(2.0) | 76.7(2.0) | 72.6(3.0) | 61.6(3.8) | 45.9(5.7) | 72.9(2.0) | 61.6(5.2) | 64.5(2.2) |
| 3 | **80.8(1.7)** | **80.9(1.7)** | 79.4(1.3) | **80.5(1.2)** | 79.3(2.2) | 58.6(3.8) | 49.2(9.7) | 76.1(2.6) | 60.2(4.9) | 61.0(2.9) |
| 4 | **84.4(2.3)** | **84.1(1.7)** | 78.2(3.0) | 80.0(2.2) | 78.0(4.5) | 47.0(7.6) | 64.0(13.) | 79.7(2.9) | 49.5(12.) | 59.2(3.3) |
| 5 | **84.1(2.8)** | **84.3(2.9)** | 77.2(2.9) | 81.4(2.5) | 77.6(4.9) | 51.4(7.6) | 55.6(11.) | 74.5(7.9) | 44.3(6.4) | 55.5(2.3) |
| 25 | 61.8(2.4) | 62.8(3.6) | 55.6(4.4) | 54.2(3.0) | 60.5(6.0) | **67.3(1.9)** | 56.7(1.0) | 51.1(1.4) | 57.9(2.2) | 60.2(2.4) |
| 26 | **63.6(1.7)** | 62.4(2.7) | **63.6(2.0)** | **62.6(1.7)** | 61.9(2.3) | 48.3(3.2) | **63.0(2.4)** | **63.7(1.4)** | **61.9(4.2)** | 59.9(3.2) |
| 27 | 60.2(2.4) | 60.9(5.2) | 59.7(3.2) | 59.3(2.7) | 59.5(3.1) | 57.4(6.0) | **68.3(2.6)** | 64.9(1.9) | 61.1(2.3) | 62.2(2.4) |
| 28 | **71.6(4.7)** | 70.8(3.9) | 63.7(5.1) | 67.4(3.7) | 66.5(5.7) | 67.5(3.1) | **72.5(1.8)** | 66.3(3.2) | **72.1(3.3)** | 69.0(2.3) |
| 29 | 55.5(3.1) | 55.3(3.4) | 54.9(2.4) | 55.9(1.9) | 56.7(2.5) | 50.6(5.3) | **59.6(2.9)** | 57.9(2.2) | 55.8(4.2) | **57.5(1.9)** |
| Avg | **72.4(11.)** | **72.4(11.)** | 68.4(9.9) | 69.9(11.) | 69.1(9.4) | 56.0(8.8) | 57.3(12.) | 68.5(9.5) | 58.0(9.0) | 61.0(4.6) |

on all datasets. In Tables 1 − 6, the boldface denotes the best and comparable methods according to the paired t-test at the significance level of 5%. ProT showed the best/comparable AUCs in almost all target domains (25 of 29 cases) and ProS also showed the best/comparable AUCs in many target domains (21 of 29 cases). Supervised anomaly detection methods (NN, NNAUC, and AEAUC) tended to perform better than semi-supervised methods (AE and OSVM) for all datasets, which suggests the effectiveness of using anomaly labels on the related domains. ProT and ProS outperformed these supervised methods. Moreover, ProT and ProS outperformed the transfer learning methods (CCSA, TOSVM, and OTL) by modeling the domain difference via the latent domain vectors. As for ProT and ProS, ProT showed better results than ProS in MNST-r and Anuran Calls since ProT uses target normal instances for training. Although ProS does not use target normal instances for training, it performed almost the same as ProT in Landmine and IoT, which indicates the effectiveness of inferring domain-specific anomaly detectors without re-training. Overall, these results showed that the proposed method detects anomalies superiorly on the target domains.

Third, we investigated the effect of considering the uncertainty of the latent domain vectors in the proposed method. To asses this, we consider the deterministic variants of ProT and ProS, called D-ProT and D-ProS, respectively. The objective function for both methods is obtained by replacing Eq. (2) with the delta distribution $q_\phi(\mathbf{z}_d | \mathbf{X}_d^-) = \delta(\mathbf{z}_d - \mu_\phi(\mathbf{X}_d^-))$ and omitting the KL divergence terms in Eqs. (4) and (5). Note that these methods also do not exist in previous studies, and thus, we can regard them as our proposal. Table 5 shows the average AUCs over all target domains of each dataset. ProT and ProS performed better than D-ProT and D-ProS in all datasets. These results show the effectiveness of considering the uncertainty in the proposed method.

Fourth, we investigated the effect of the number of anomalous training instances. Table 6 shows the average AUCs over target domains when the anomaly rate of each source domain $r_a := N_d^+/(N_d^+ + N_d^-)$ was equally changed on MNIST-r. As expected, as the number of anomalous training instances decreased, ProT, ProS, and AEAUC came to perform worse. However, even if the anomaly rate $r_a$ is small, ProT and ProS showed better results than AE. In addition, ProT and ProS outperformed AEAUC for all the values of the $r_a$. This result suggests that the proposed method is relatively robust against the anomaly ratio $r_a$.

Last, we evaluated the computation time of the proposed method. We evaluated the training time of 100 epochs for ProT, ProS, and AEAUC on MNIST-r. We set the hyperparameters as follows: the regularization parameter of the AUC loss $\lambda$ was $10^4$, the dimension of the latent domain vector $K$ was 20, the regularization parameter of the latent domain vector $\beta$ was one, and the sample size of the reparametrization trick $L$ was one. The computation time of 100 epoch training of ProT, ProS, and AEAUC were 10.58, 9.94 and 5.56 seconds, respectively. Although ProT and ProS took more computation time than AEAUC due to the additional network for the latent domain vector, the computation costs of ProT and ProS were not so large. Since ProT uses target normal instances

Table 4: Average and standard deviation of AUCs [%] on each target domain on IoT.

| Target | ProT | ProS | NN | NNAUC | AEAUC | AE | OSVM | CCSA | TOSVM | OTL |
|---|---|---|---|---|---|---|---|---|---|---|
| Dbell | **99.6(0.1)** | **99.6(0.1)** | **99.4(0.4)** | 99.5(0.2) | **99.5(0.1)** | 82.1(13.) | 99.1(0.2) | **99.3(0.4)** | **99.5(0.2)** | 99.2(0.2) |
| Therm | **99.6(0.1)** | **99.6(0.1)** | **99.6(0.1)** | **99.6(0.1)** | **99.5(0.2)** | 90.8(11.) | 98.6(0.2) | **99.6(0.1)** | **99.6(0.1)** | 97.8(0.4) |
| Ebell | **99.6(0.2)** | **99.6(0.1)** | **99.5(0.1)** | **99.6(0.1)** | **99.6(0.2)** | 92.8(6.6) | 97.4(0.3) | **99.5(0.2)** | **98.4(2.0)** | 97.5(0.4) |
| Baby | 94.3(1.3) | **94.7(1.0)** | 93.1(1.4) | 93.5(1.3) | 90.4(2.7) | 46.8(3.3) | 69.3(0.8) | 94.1(1.4) | 80.8(9.1) | 92.3(2.3) |
| 737 | **98.5(0.4)** | 98.3(0.6) | 97.9(0.6) | 98.1(0.5) | **98.1(1.0)** | 68.9(11.) | 89.0(1.9) | 97.5(0.5) | **97.4(2.0)** | **97.3(0.5)** |
| 838 | **99.1(0.2)** | **99.2(0.2)** | 99.0(0.3) | **99.1(0.3)** | **99.1(0.2)** | 68.4(12.) | 87.4(2.7) | **99.1(0.3)** | 96.8(2.9) | 97.9(0.4) |
| Web | **99.1(0.2)** | 99.0(0.2) | **99.2(0.1)** | **99.2(0.1)** | **99.1(0.2)** | 85.6(12.) | 97.7(0.3) | **99.1(0.1)** | **98.8(0.6)** | 97.7(0.2) |
| 1002 | **97.8(0.3)** | **97.8(0.3)** | 97.5(0.5) | **97.7(0.2)** | **97.6(0.6)** | 56.5(15.) | 72.6(2.1) | **97.7(0.2)** | 97.3(0.4) | 94.8(2.8) |
| Avg | **98.4(1.7)** | **98.5(1.6)** | 98.2(2.1) | 98.3(2.1) | 97.9(3.1) | 74.0(19.) | 88.9(11.) | 98.2(1.8) | 96.1(6.8) | 96.8(2.4) |

Table 5: The effect of considering the uncertainty. Average and standard deviation of AUCs [%] over all target domains of each dataset.

| Data | ProT | ProS | D-ProT | D-ProS |
|---|---|---|---|---|
| MNIST-r | **96.6(3.2)** | 96.0(3.6) | 95.9(4.0) | 95.5(4.0) |
| Anuran Calls | **99.8(0.6)** | 96.8(3.8) | 86.5(2.2) | 84.8(2.2) |
| Landmine | **72.4(11.)** | **72.4(11.)** | 71.1(11.) | 71.1(11.) |
| IoT | **98.4(1.7)** | **98.5(1.6)** | 98.1(2.6) | 97.9(2.8) |

Table 6: Average and standard deviation of AUCs [%] over all target domains when changing the anomaly rate $r_a$ on MNIST-r.

| $r_a$ | ProT | ProS | AEAUC | AE |
|---|---|---|---|---|
| 0.1 | **96.6(3.2)** | 96.0(3.6) | 94.3(4.9) | 72.7(3.0) |
| 0.05 | **94.5(5.0)** | **94.1(5.6)** | 91.3(7.3) | 72.7(3.0) |
| 0.01 | **89.7(7.1)** | 88.7(8.5) | 86.4(8.2) | 72.7(3.0) |
| 0.005 | **87.0(8.4)** | 85.0(9.8) | 81.9(10.) | 72.7(3.0) |

as well as source instances to learn the target-specific anomaly score function, ProT took a little more computation time than ProS. ProS infer the target-specific anomaly score function using the set of target normal instances without re-training. In this experiment, ProS inferred it with 0.0027 seconds when the sample size of the reparametrization trick $L$ was ten. This inference time was 3918 times faster than the training time of ProT. Additional experimental results such as the $\lambda$, $K$, and $\beta$'s dependency are described in the supplemental material.

# 6 Conclusion

In this paper, we proposed a method to improve the anomaly detection performance on target domains by inferring their latent domain vectors. The proposed method can infer the anomaly detectors for any domains given the sets of normal instances in the domains without re-training or anomalies. In addition, the proposed method can also use target normal instances for training. The most attractive point of the proposed method is that it can infer domain-specific anomaly detectors in two situations, i.e., target normal instances can or cannot be used for training, in a unified framework. In experiments using one synthetic and four real-world datasets, the proposed method outperformed various existing anomaly detection methods. For future work, we will apply sophisticated density models such as autoregressive models and flow-based models as the anomaly score function.

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
