[Supplementary Material · neurips_2019_supplimental_camera_ready_v1.pdf]

# Supplemental Material: Transfer Anomaly Detection by Inferring Latent Domain Representations

**Atsutoshi Kumagai**
NTT Software Innovation Center
NTT Secure Platform Laboratories
atsutoshi.kumagai.ht@hco.ntt.co.jp

**Tomoharu Iwata**
NTT Communication Science Laboratories
tomoharu.iwata.gy@hco.ntt.co.jp

**Yasuhiro Fujiwara**
NTT Communication Science Laboratories
yasuhiro.fujiwara.kh@hco.ntt.co.jp

## 1 Datasets Details

The synthetic dataset consists of eight double circles (domains) around the $(0, 0)$. Each double circle has outer and inner circles that consist of normal and anomalous instances, respectively. Each instance of the outer circle was generated from the point on the unit circumference by adding the Gaussian noise with the $(0.1)^2$ variance. Each instance of the inner circle was generated from the point on the circumference of a 0.05 radius by adding the Gaussian noise with the $(0.1)^2$ variance. After this generation, the center of each double circle was arranged at $45$-degree intervals on the unit circumference to create eight different double circles. We used the '7'-th domain as the target domain and the rest as the source domains. In each source domain used for training, we used 1,050 (50 anomalous and 1,000 normal) instances for training, and 210 (10 anomalous and 200 normal) instances for validation.

The MNIST-r[1] is derived from the MNIST, which was introduced by Ghifary et al. [3]. Each domain is created by rotating the images in multiples of 15 degrees: 0,15,30,45,60, and 75. Each domain has 1,000 images, which are represented by 256-dimensional vectors, of 10 classes (digits). We selected the '4' digit as the anomalous class and the rest as the normal class. The average anomaly rate $N_d^+/(N_d^+ + N_d^-)$ is 0.1. In each source domain used for training, we randomly selected 80% of instances for training and 20% for validation.

The Anuran Calls[2] is a real-world dataset collected from frog croaking sounds, which is used in the multi-task anomaly detection study [4]. We picked five major species of *AdenomeraAndre (Ade)*, *Ameeregatrivittata (Ame)*, *HylaMinuta (Hyl)*, *HypsiboasCinerascens (HCi)*, and *HypsiboasCordobae (Hco)*, as domains, in which the first ten MFCCs features were used the same as Idé et al. [4]. We used the species of *Rhinellagranulosa* as the anomalous instances the same as Idé et al [4] and assigned 13 anomalous instances to each domain. Each domain has different anomalies. As a result, each domain has 686 instances on average, and the average anomaly rate $N_d^+/(N_d^+ + N_d^-)$ is 0.024. In each anomalous/normal class of source domain used for training, we randomly selected 70% of instances for training and 30% for validation.

The Landmine[3] is a real-world dataset which is well-used in multi-task learning studies [10]. It consists of 29 domains, and each instance is a nine-dimensional vector extracted from radar image that captures a region of landmine fields. Each domain has 511 instances on average, and the average anomaly rate $N_d^+/(N_d^+ + N_d^-)$ is 0.062. Among the 29 domains, domains 1–15 correspond to

regions that are relatively highly foliated while the other domains 16–29 correspond to regions that are bare earth or desert. We used ten domains that consist of the first five (1-5) and the last five (25-29) domains. We normalized each feature vector by $\ell_2$-normalization. In each source domain used for training, we randomly selected 70% of instances for training and 30% for validation.

The IoT[4] contains real network traffic data, which are gathered from nine IoT devices infected by malware. The IoT has normal and malicious traffic instances. Each instance is a 115-dimensional vector and was normalized by $\ell_2$-normalization. We used the devices that had normal data. We treated each device as a domain and randomly choose instances from the first 5,000 traffic data from each normal/attack file of each device file. As a result, for each source domain for training, we used 1,000 (50 anomalous and 950 normal) instances for training, 100 (5 anomalous and 95 normal) for validation.

For MNIST-r, Anuran Calls, and Landmine, we randomly selected 60% of normal instances for training, 10% of normal instances for validation, and 30% of normal instances and the remaining anomalous instances for testing from each target domain. For the synthetic dataset, we used 1,000 normal instances for training, 200 normal instances for validation, and 1,000 normal and 50 anomalous instances for testing from the target domain. For IoT, we randomly selected 950 normal instances for training, 50 normal instances for validation, and 1,900 normal and 1,900 anomalous instances for testing from each target domain. Note that validation normal instances in the target domain were used for the AE.

## 2 Experimental Setup Details

### 2.1 Proposed Method Variants

We evaluated two variants of the proposed method: ProT and ProS. ProT uses normal instances in the target domain as well as anomalous and normal instances in the source domains for training. ProS does not use normal instances in the target domain for training. Instead, ProS infers the anomaly score function of the target domain using the set of training normal instances in the domain. For the AE networks of both methods, the encoder was a feed-forward neural network with one-hidden layer with 100 hidden nodes and rectified linear unit (ReLU) activations. Similarly, the decoder was a feed-forward neural network with one-hidden layer with 100 hidden nodes and the ReLU activation. For MNIST-r, sigmoid function was used for the output of the decoder. For Anuran Calls, Landmine and IoT, the tanh function was used for the output of the decoder. The output sizes of the encoder were 1, 20, 5, 5, and 20 for Synthetic, MNIST-r, Anuran Calls, Landmine, and IoT, respectively. For $\mu_\phi(\mathbf{X}_d^-)$ and $\ln \sigma_\phi^2(\mathbf{X}_d^-)$ of $q_\phi(\mathbf{z}_d|\mathbf{X}_d^-)$, we used feed-forward neural networks with one-hidden layers. Specifically, the shared single-layer neural network with 100 nodes and ReLU activations was used as $\eta$, and different single-layer neural networks were used as $\rho$. We took average of $\eta(\mathbf{x}_{dn}^-)$ before applying $\rho$ to reduce the data size difference of each domain. That is, we used $\tau(\mathbf{X}_d^-) = \rho\left(\frac{1}{N_d^-}\sum_{n=1}^{N_d^-}\eta(\mathbf{x}_{dn}^-)\right)$ as neural networks for inferring the latent domain vectors, $\mu_\phi(\mathbf{X}_d^-)$ and $\ln \sigma_\phi^2(\mathbf{X}_d^-)$. Note that these neural networks are also included in the definition of the deep sets [11]. The latent domain vector $\mathbf{z}_d$ was concatenated with the input $\mathbf{x}_d$ for the anomaly score functions.

### 2.2 Comparison Methods

We compared the proposed method with eight methods: NN, NNAUC, AEAUC, AE, OSVM, CCSA, TOSVM, and OTL.

**AE** is the autoencoder [8]. We used the same network architecture as the proposed method. Specifically, we used the two feed-forward neural networks with one-hidden layers with 100 hidden nodes and ReLU activations as the encoder and the decoder, respectively. The output function for the decoder and the number of the encoder outputs were the same as that of the proposed method.

**OSVM** is the one-class support vector machine [9], which is an extension of the support vector machine to one-class classification. We used the RBF kernel as a kernel function of OSVM.

**NN** is the feed-forward neural network classifier. We used a feed-forward neural network with one-hidden layer with 100 hidden nodes, ReLU activations, and a softmax output function.

**NNAUC** is the feed-forward neural network classifier for class-imbalanced data. NNAUC uses the differentiable AUC loss instead of the cross entropy loss in NN to deal with the class-imbalance.

**AEAUC** is the autoencoder based classifier for class-imbalanced data [5]. The loss function of AEAUC consists of two terms: the anomaly score of normal instances and the differentiable AUC regularizer. We used the reconstruction error as the score function for a fair comparison. AEAUC is obtained from the proposed method on the source domains by omitting the latent domain vectors.

**CCSA** is a recently proposed method for semi-supervised domain adaptation [7]. CCSA learns domain-invariant features by introducing the contrastive semantic alignment loss, which brings instances with the same labels closer and separates instances with different labels in the hidden space. CCSA can be used when the target domain has only the normal class [7]. CCSA used NN as the base neural network.

**TOSVM** is the transfer one-class support vector machine [1]. This method belongs to the two-step approach for transfer anomaly detection. TOSVM first learn features using anomalous and normal instances in source domains, and then perform OSVM using the normal instances in the target domain with the pre-trained features. We used the hidden layer of NN as the pre-trained features.

**OTL** is the one-class data transfer learning [2]. OTL predicts anomalous class distributions from normal class distributions in the target domains using target normal instances and anomalous and normal source instances. OTL learns classifiers for the target domain with the estimated distribution.

## 2.3 Hyperparameters

We selected hyperparameters using average validation AUC on the source domains for all methods except for AE. AE used the validation reconstruction error on the target domains. For CCSA, the regularization parameter for contrastive semantic alignment loss $\gamma$ was chosen from $\{0.01, 0.1, 0.25, 0.5\}$. For OSVM and TOSVM, the kernel parameters was selected from $\{10^{-3}, 10^{-2}, \ldots, 10^3\}$. For OTL, the number of the weak classifiers was chosen from $\{10, 50, 100, 500, 1000\}$. For the proposed method and AEAUC, the regularization parameter $\lambda$ was chosen from $\{0, 0.1, 1, 10, \ldots, 10^4\}$. The dimensionality of the latent domain vector $K$ of the proposed method was chosen from $\{1, 5, 10, 20\}$, and the sample size of the reparametrization trick $L$ was set to one for training and ten for testing. We set the weight parameters $\alpha_d$ as one for all domains used for training, and the regularization parameter for the latent domain vectors $\beta = 1$ for all datasets. We used the Adam optimizer [6] with a learning rate of $0.001$. The maximum numbers of epochs were 200, 500, 1000, 300, and 200 for Synthetic, MNIST-r, Anuran Calls, Landmine, and IoT, respectively. We evaluated the test AUC when the method obtained the best validation AUC after 15 epochs to avoid over-fitting. We conducted experiments on ten different datasets for each target domain and reported the mean test AUC.

## 3 Additional Experimental Results

### 3.1 Results for Another Synthetic Dataset

To obtain further insights, we investigated how the proposed method works using another two-dimensional synthetic dataset, called Synthetic2.

Figure 1a represents Synthetic2, which consists of five aligned double circles (domains). Each double circle (domain) has outer and inner circles that consist of normal and anomalous instances, respectively. The $d + 1$-th domain was created by shifting $d$-th domain $2.5$ in the $x$-axis direction. We used the '4'-th domain as the target domain and the rest as the source domains. In each source domain used for training, we used 1,050 (50 anomalous and 1,000 normal) instances for training, and 210 (10 anomalous and 200 normal) for validation. For the target domain, we used 1,000 normal instances for training, 200 normal instances for validation, and 1,050 (50 anomalous and 1,000 normal) instances for testing. The validation normal instances in the target domain were used for the AE.

|(a) Synthetic2 | (b) Latent Domain Vectors | (c) Anomaly Score | (d) AUC |

Figure 1: (a) Synthetic2 consists of five aligned double circles (domains). Anomalous and normal source instances are represented by orange and blue points, respectively. Anomalous and normal target instances are represented by red and green points, respectively. (b) Posteriors of the latent domain vectors estimated by ProS. Each point denotes the mean of each posterior. (c) Heatmap of anomaly scores on the target domain obtained by ProS. Darker color indicates higher anomaly score. (d) Average and standard error of AUCs when $\lambda$ was changed.

|(a) MNIST-r | (b) Anuran Calls | (c) Landmine | (d) IoT |

Figure 2: Average AUCs over the target domains of each dataset when $\lambda$ was changed. The $x$-axis denotes $\lambda$ and the $y$-axis denotes AUC.

Figure 1b represents an example of the posterior of the latent domain vectors estimated by ProS with $K = 2$. We found that the estimated latent domain vectors were consistently aligned in order. This indicates that the proposed method was able to capture the property of domains (parallel translation) as the latent domain vectors. Figure 1c shows an example of the heatmap of anomaly scores on the target domain obtained by ProS. We found that ProS was able to give high (low) anomaly scores to the anomalous (normal) region on the target domain. Figure 1d shows average and standard deviation of AUCs on the target domain when changing the value of $\lambda$. We confirmed that ProT and ProS obviously outperformed the others when the values of $\lambda$ were relatively large.

## 3.2 Dependency of the Regularization Weight for the AUC loss $\lambda$

Figure 2 showed the average and standard error of AUCs over all target domains of each real-world dataset when changing the regularization weight $\lambda$ of the AUC term. We used AEAUC as a baseline since it is obtained from ProS by omitting the latent domain vectors. The best $\lambda$ for ProT and ProS differed across datasets. We found that ProT constantly outperformed the others for all datasets except Landmine when $\lambda$ was changed. For Landmine, ProT showed better results than the others when $\lambda$ was not small. Similarly, ProS constantly outperformed AEAUC for MNIST-r and IoT and showed better results when $\lambda$ was relatively large for Landmine. For Anuran Calls, ProS deteriorated the performance compared with AEAUC except for $\lambda = 1$. However, even Anuran Calls, we confirmed that ProS was able to select $\lambda = 1$ safely by using validation data on the source domains.

## 3.3 Dependency of the Dimension of the Latent Domain Vector $K$

Figure 3 showed the average and standard error of AUCs over all target domains of each real-world dataset when changing the dimension of the latent domain vectors $K$. For MNIST-r and Anuran Calls, ProT constantly performed better than ProS for all $K$. For Landmine and IoT, ProT and ProS performed almost the same for all $K$. For MNIST-r and Landmine, ProT and ProS performed better as the value of $K$ increased. In contrast, ProT and ProS showed the best AUCs when $K = 1$ for Anuran Calls. Although the optimal $K$ depends on datasets, we were able to obtain it using validation data on the source domains.

| (a) MNIST-r | (b) Anuran Calls | (c) Landmine | (d) IoT |

Figure 3: Average AUCs over the target domains of each dataset when $K$ was changed. The $x$-axis denotes $K$ and the $y$-axis denotes AUC.

| (a) MNIST-r | (b) Anuran Calls | (c) Landmine | (d) IoT |

Figure 4: Average AUCs over the target domains of each dataset when $\beta$ was changed. The $x$-axis denotes $\beta$ and the $y$-axis denotes AUC.

### 3.4 Dependency of the Regularization Weight for the Latent Domain Vectors $\beta$

Figure 4 showed the average and standard error of AUCs over all target domains of each real-world dataset when changing the value of the regularization weight for the latent domain vectors $\beta$. For MNIST-r and Anuran Calls, ProT constantly performed better than ProS for all $\beta$. For Landmine and IoT, ProT and ProS performed almost the same for all $\beta$. Although the best $\beta$ for ProT and ProS differed across datasets, both methods were relatively robust against the values of $\beta$.

## Footnotes

[1]https://github.com/ghif/mtae

[2]https://archive.ics.uci.edu/ml/datasets/Anuran+Calls+%28MFCCs%29

[3]http://people.ee.duke.edu/ lcarin/LandmineData.zip

[4]https://archive.ics.uci.edu/ml/datasets/detection_of_IoT_botnet_attacks_N_BaIoT