[Reviews · NeurIPS 2019]

Reviewer 1



This paper presents a novel approach to transfer learned representations to new but related domains. Detectors parameters for new domains are inferred from existing ones without re-training. This approach is well suited to anomaly detection where anomaly labels are hard to obtain because of the rarity of anomalies. Additionally the method is flexible as to allow to target domain data to be used for training. The anomaly score function parameters for a target domain are inferred from the trained parameters (which may or may not include target domain instances) and latent domain vectors which are estimated by sampling from a model (multivariate gaussian or other). Experiments include a synthetic toyset as well as benchmark datasets including rotated MNIST. In the case of MNIST the domains are rotations of the digits by 15 degrees increments (6 domains are created: 0 to 75). Five domains are used for training and the remaining for transfer. The method is compared to 8 other approaches for anomaly detection. Results demonstrate that the method works very well and surpasses all other methods. PROS: The paper is well written, technically sound and the mathematical exposition is quite extensive. The method is original and novel. The experiments are relatively extensive and the results show that the approaches surpasses other anomaly detection approaches. The method is flexible in that it allows for normal instance of the target domain to be used if present at training time, but also allows for the anomaly score function to be inferred when those instance are not present. CONS: The author should give some intuition as to when the approach starts breaking apart when the source and target domains become less related. The toy dataset and MNIST-r exhibit clear domains, however, the other datasets' domains are not intuitive. The computational complexity of the method is only briefly stated without any justification. Given that sampling may sometimes be expensive, it would be useful to get a feel for it. In particular since IoT applications are mentioned that would benefit from not needing to retrain, it would be interesting to see the speedups. A pseudo-code algorithm would be helpful for somebody wanting to reproduce these experiments. For MNISR-r, the author do not specify what the domains are. One can guess from table 1 that they are 15-degrees rotation increments.

Reviewer 2



This paper proposed a transfer learning based anomaly detection algorithm to infer the anomaly scores of the target domain examples by transferring knowledge on related domains. To achieve this, a latent domain representation vectors are trained to capture the domain knowledge. Quantitative experiments and empirical studies are presented to manifest the efficacy of the proposed algorithm. In general, the paper is well-written and well-organized. My main concern is that there lacks some insightful discussion regarding the proposed problem and algorithm. In particular, (1) The proposed algorithm requires to obtain the exact number of normal/abnormal examples in each domain. However, it could be impractical in real-world applications. (2) It is often the case that the target domain and source domain may have the imbalanced sizes of data. Will it be problematic for the proposed algorithm? (3) In many applications, one may observe that the anomalies are partially overlapped with the normal patterns. Can the proposed algorithm handle such specific case?

Reviewer 3



This paper proposed a practical method for anomaly detection in different domain. The proposed method does not require anomalous instances and could be used directly or fine-tuned in the target domain. The authors present their idea logically. For example, they combine Gaussian distribution and Neural networks to generate latent vector; They add a regularization term to avoid overfitting and modifies loss function to use only normal instances in finetuning. The experiments on 4 public datasets verify the effectiveness of the proposed method. The experiments in section 5 and supplementary material are complete. However, the 4 datasets are relatively simpler comparing to CV datasets. And even the method NN(neural network classifier with only one hidden layer) could achieve a good result on the 4 datasets. Could the proposed method be used on CV datasets? Is the proposed method limited by the performance of encoder & decoder? Besides, for section 4.2, how the Z works in network F and how the Z is estimated by neural network are not clearly indicated. That would be important details to reproduce the method.

[Author Response · NeurIPS 2019]

Table 1: Average of AUCs [%] over all target domains of each dataset.

| Dataset | ProT | ProS | AEAUC | ProT($\lambda=0$) | ProS($\lambda=0$) | AEAUC($\lambda=0$) | ProT($\beta=0$) | ProS($\beta=0$) |
|---|---|---|---|---|---|---|---|---|
| MNIST-r | **96.6(3.2)** | 96.0(3.6) | 94.3(4.9) | 70.9(4.4) | 64.1(4.8) | 63.2(4.9) | 95.8(4.0) | 95.5(4.2) |
| Anuran Calls | **99.8(0.6)** | 96.8(3.8) | 96.9(4.0) | 71.3(15.) | 28.8(16.) | 28.1(16.) | **97.9(8.7)** | 91.9(11.) |
| Landmine | **72.4(11.)** | **72.4(11.)** | 69.1(9.4) | 55.3(8.1) | 55.4(9.3) | 54.4(8.8) | **72.1(11.)** | **71.8(11.)** |
| IoT | 98.4(1.7) | **98.5(1.6)** | 97.9(3.1) | 84.6(8.5) | 79.9(9.9) | 77.3(13.) | **98.5(1.4)** | **98.5(1.5)** |

We would like to thank the reviewers for their feedback and insightful comments, which we shall address below.

**Reviewer #1**

**>For MNISR-r, the author...:** As you mentioned, the domains of MNIST-r correspond to 15-degree rotation incre-
ments, which is described in the supplemental material (L12-13). We will specify this in the revised main paper. **5.**
**Improvements 1):** We will give some intuition about our approach. Our method might not be effective when the source
and target domains become less related. However, our method would reduce the negative effects of this irrelevance since
it considers the uncertainty of the latent domain vectors. That is, when the distributions of the source and target normal
instances differ greatly or there is a small amount of target instances, the latent domain vector of the target domain
would have large variance $\sigma^2_\phi(\mathbf{X}_d^-)$. This variance alleviates the negative transfer since it prevents over-fitting. When
target normal instances are available for training, our method would future reduce the negative transfer since the scores
of target normal instances are directly learned to become low. We will investigate this. **2):** We will add the detailed
explanation of the computational complexity for the inference described in Sec. 4.4. To infer the parameters $\mu_{\phi_*}(\mathbf{X}_{d'}^-)$
and $\ln \sigma_{\phi_*}(\mathbf{X}_{d'}^-)$ from $\mathbf{X}_{d'}^-$, our method requires $N_{d'}^-$ feed-forward passes of the instances in the neural networks with
the parameter $\phi_*$. Besides, to sample $L$ latent domain vectors, our method requires $L$ samplings from the standard
Gaussian distribution. Note that sampling from the standard Gaussian distribution is lightweight. As a result, the total
computation complexity for the inference of the anomaly score function $s(\cdot)$ becomes $\mathcal{O}(N_{d'}^- + L)$. We will clarify this
in the revised paper. We evaluated the inference time of ProS on IoT. The experimental setup is the same as that in Sec.
3.5 of the supplemental material. The inference time of the $s(\cdot)$ with $L = 10$ was 5708 times faster than the training
time of ProT for 100 epochs (0.0025 sec. vs 14.27 sec.) without losing the detection performance. This result shows the
benefit of not needing to retrain. **3):** Thank you for your suggestion. We will add the pseudo-code in the revised paper.

**Reviewer #2**

**2. Detailed comments (1):** Our method does not require the exact number of normal/anomalous instances. Our method
can use unlabeled instances, which are unknown whether anomalous or normal, by treating them as normal instances
assuming most of the unlabeled instances are normal. This technique is commonly used in unsupervised anomaly
detection methods [6,8]. **(2):** Our method would reduce the harmful effects of the data size difference of each domain
since the objective function for each domain Eq. (3) is normalized by the data size. As to the neural networks for the
latent domain vectors, we can also reduce these effects by taking average of $\eta(\mathbf{x}_{dn})$ as described in the supplemental
material (L64-68). Indeed, our method with this architecture worked well against the imbalanced size datasets (Anuran
Calls and Landmine) in the experiment. **(3):** Our method can detect anomalies that are partially overlapped with normal
instances when a large value is set to the regularization parameter $\lambda$ in Eq. (3) since training anomalies are well learned
as the larger value $\lambda$. However, in this case, there is a risk that false detection of overlapped normal instances increases.
In practice, we would select the appropriate value of $\lambda$ using validation data on the source domains.

**Reviewer #3**

**1. Contributions 1):** Although we used AEs as the anomaly score functions due to their simplicity and effectiveness,
which have been described in many studies [6,43,56,12,1], the proposed framework can use other semi-supervised
anomaly detection methods with learnable parameters such as autoregressive models [19,46] and flow-based models
[15,31] as described in Introduction and Sec. 4 (L62-66 and L159-161). Thus, our framework is not limited to the
performance of AEs. **5. Improvements 1):** Our method can be applied to CV datasets with CNN-based AEs. We will
evaluate it. **2):** We used the neural network with the parameter $\phi$ to represent each domain (instance set) as the latent
domain vector. This neural network can preserve all the properties of the instance set with suitable $\rho$ and $\eta$ as described
in [54]. The parameter $\phi$, which is shared among all domains, is learned so as to infer $\mathbf{z}_d$'s that can detect anomalies
well in each domain by maximizing the objective function $\mathcal{L}$. The $\mathbf{z}_d$ modifies the hidden states of the original AE so
that anomalies cannot be reconstructed (high anomaly score) but normal instances can (low anomaly score) in each
domain. We will clarify this in the revised paper. **3):** We conducted two additional experiments. Specifically, we
evaluated our method without the regularization ($\beta = 0$ in Eqs. (4) and (5)) and it without the anomalous instances
($\lambda = 0$ in Eq. (3)). Tables 1 in this response shows the results. ProT and ProS with $\lambda = 0$ obviously deteriorated
the performance, which means the importance of using anomalies in the related domains. ProT and ProS with $\beta = 0$
showed similar performance with $\beta \neq 0$ in Landmine and IoT, but worse performance in MNIST-r and Anuran Calls,
which suggests the efficacy of the regularization. We will add these results in the revised paper.

[Meta-Review · NeurIPS 2019]

This is a mixed paper. In terms of problem setting and methodology, the proposed problem is interesting, and the proposed method is novel. In terms of experiments, there are some concerns on the experimental setup. After rebuttal, two reviewers still have concerns on the practical issues of the proposed method on real-world scenarios. One is not convinced by authors' response on how to apply the proposed method to handle the special case, where the anomalies are partially overlapped with the normal patterns. The other is not satisfied that the authors do not conduct additional experiments on larger CV datasets, who even lowers the overall score from 6 to 4. After reading through this paper, I find that the proposed idea is interesting and novel, which does provide some new insights to the field of transfer learning in the application to abnormal detection. I believe this is more important and useful than achieving SOTA results by developing incremental methods. Though there may be some practical issues when applying the proposed method to real-world complex scenarios as mentioned by reviewers, the proposed method looks reasonable to most general cases of cross-domain anomaly detection. Of course, if the authors can address the issues raised by the reviewers, e.g., conducting more experiments on larger CV datasets (though I do not think it is a must), and address the overlapping issue of anomalies and normal patterns, then they can make this work stronger. Overall, by considering the technical novelty, I think this work is still deserved to be published in NeurIPS.